# PTEN, A Target of Microrna-374b, Contributes to the Radiosensitivity of Canine Oral Melanoma Cells

**DOI:** 10.3390/ijms20184631

**Published:** 2019-09-18

**Authors:** Shunsuke Noguchi, Ryo Ogusu, Yusuke Wada, Satoshi Matsuyama, Takashi Mori

**Affiliations:** 1Laboratory of Veterinary Radiology, Graduate School of Life and Environmental Sciences, Osaka Prefecture University, 1-58 Rinku Ourai Kita, Izumisano-shi, Osaka 598-8531, Japan; ryo.hand.2148.ro.ev@gmail.com (R.O.); matuyama@vet.osakafu-u.ac.jp (S.M.); 2Veterinary Medical Center, Osaka Prefecture University, 1-58 Rinku Ourai Kita, Izumisano-shi, Osaka 598-8531, Japan; y-wada@vet.osakafu-u.ac.jp; 3Laboratory of Veterinary Clinical Oncology, Joint Department of Veterinary Medicine, Faculty of Applied Biological Sciences, Gifu University, 1-1 Yanagido, Gifu 501-1112, Japan; tmori@gifu-u.ac.jp

**Keywords:** dog, melanoma, miR-374b, PTEN, radiation therapy

## Abstract

Canine oral malignant melanoma (CoMM) is often treated by radiation therapy in veterinary medicine. However, not all cases are successfully managed by this treatment. For improved efficacy of radiation therapy, biomarkers predicting the radiosensitivity of melanoma cells need to be explored. Here, we, first, developed the radioresistant CoMM cell line, KMeC/R. We found that the expression level of phosphatase and tensin homolog (PTEN) of KMeC/R cells was significantly downregulated compared with KMeC cells. Overexpression of PTEN successfully restored the radiosensitivity of KMeC/R cells, and silencing of PTEN significantly increased the radioresistance of the CoMM cells tested. Next, we focused on microRNAs (miRNAs) to explore the mechanisms of downregulation of PTEN in KMeC/R cells. miR-374b was upregulated in KMeC/R cells compared with that in KMeC cells and in the irradiated CoMM cells tested. Furthermore, miR-374b directly targeted PTEN based on the luciferase activity assay. Moreover, the extrinsic miR-374b significantly increased the radioresistance of KMeC cells. In addition, the expression level of PTEN was significantly downregulated and that of miR-374b tended to be upregulated in recurrent CoMM tissues after radiation therapy compared with the pre-treatment tissues. Thus, the current study suggested that the miR-374b/PTEN signaling pathway possibly plays an important role in CoMM radiosensitivity.

## 1. Introduction

Melanoma is the most common malignant tumor of the canine oral cavity [1]. Canine melanoma has been considered to be important as a preclinical model of human melanoma based on the molecular analysis [2,3]. Radiation therapy (RT) is often applied to canine oral malignant melanoma (CoMM), although human melanoma is rarely so treated by RT as a first-line therapy due to radioresistance. In their recent study, Kawabe et al. suggested that the therapeutic efficacy of RT is almost equivalent to that of surgical treatment for CoMM [4]. However, local control of it is sometimes unsuccessful due to no response to RT or to early recurrence. Therefore, biomarkers predictive of the efficacy of RT or effective radiosensitizers need to be explored to improve the therapeutic efficiency in veterinary medicine, as well as in human medicine.

PTEN, the acronym for phosphatase and tensin homolog, is well known to function as a tumor suppressor gene by negatively modulating the phosphoinositide 3-kinase (PI3K)/v-akt murine thymoma viral oncogene homolog (Akt)/mammalian target of the rapamycin (mTOR) signaling pathway [5]. It has been reported that downregulation of PTEN, which is targeted by miR-222 and -29b, is associated with radioresistance in human nasopharyngeal carcinoma and cervical cancer [6,7]. In addition, PTEN is epigenetically silenced in human melanoma [8]. On the other hand, the differential expression and role of PTEN remains unclear in CoMM, although loss or downregulation of PTEN has been reported in several kinds of canine cancer such as mammary tumor and prostate carcinoma [9,10].

microRNAs (miRNAs) are functional non-coding RNAs and are known to contribute to cancer development and progression in both humans and canines, as numerous studies have clarified [11,12]. Moreover, miRNAs have been reported to contribute to radiosensitivity or radioresistance in several kinds of human cancer [6,13,14]. More recently, we reported that miR-205, which is downregulated and functions as an anti-oncomir in CoMM cells, affects radiosensitivity in CoMM cells [15]. Thus, it seems to be valuable to identify the miRNAs involved in radiosensitivity or radioresistance, which may lead to improving the therapeutic efficacy of RT for CoMM. 

The aim of this study was to explore gene and miRNAs targeting involved in radiosensitivity or radioresistance in CoMM cells and CoMM tissues.

## 2. Results

### 2.1. Development of Radioresistant CoMM Cells

First, we developed the radioresistant cell line (KMeC/R; Figure 1a). Next, we examined the PTEN expression level of KMeC and KMeC/R cells, because PTEN has been known as one of the radiosensitivity-associated proteins [6,7]. As a result, expectedly, the expression level of PTEN was markedly downregulated in KMeC/R cells compared with that in KMeC cells.

### 2.2. Overexpression of PTEN Canceled the Radioresistance of KMeC/R Cells and Silencing of PTEN Enhanced that of KMeC Cells

To validate that downregulation of PTEN was associated with the radioresistance of KMeC/R cells, we constructed a PTEN expression plasmid vector and developed siRNA for PTEN. Expectedly, overexpression of PTEN by the transfection with the PTEN expression vector completely canceled the radioresistance of KMeC/R cells (Figure 2a,b). Inversely, the knockdown of PTEN significantly promoted the radioresistance of CMM1 and KMeC cells (Figure 2c,d).

### 2.3. Differential Expression of miRNAs Associated with Irradiation

To evaluate the miRNAs showing differential expression between pre- and post-irradiation, we performed microRNA microarray using human melanoma A2058 cells. We used human melanoma cells for this experiment, because the number of mounted miRNA probes on the array was much greater for human than for canine. After irradiation at a dose of 2 Gy, which dose was almost LD_50_ of A2058 cells (data not shown), 28 miRNAs were upregulated and 27 downregulated in A2058 cells (Figure 3a). Of those miRNAs, we focused on miR-374b-5p, which was upregulated after irradiation, because miR-374b-5p potentially targets PTEN based on TargetScan (http://www.targetscan.org/vert_72/). Similar to the results of the microarray, miR-374b-5p (termed miR-374b in *Canis familiaris*) was significantly upregulated after irradiation in all of the canine melanoma cell lines tested (Figure 3b). In addition, the expression level of PTEN was consistently downregulated (Figure 3b). Moreover, the expression level of miR-374b was significantly upregulated in KMeC/R cells compared with in KMeC cells (Figure 3c). While the data indicated that the expression of miR-374b was increased in response to irradiation, the role of miR-374b was unclear. Therefore, the function of miR-374b was subsequently evaluated.

### 2.4. miR-374b-5p Reduced the Expression Level of PTEN and Conferred the Radioresistance

Next, to validate the function of miR-374b, we transfected canine melanoma cells with a miR-374b-5p mimic. As shown in Figure 4a, treatment with extrinsic miR-374b-5p decreased the expression level of PTEN in CMM1 and KMeC cells, and the effect in KMeC cells was more marked than that in CMM1 cells. Moreover, the miRNA significantly increased the resistance of KMeC cells to irradiation but not that of CMM1 cells (Figure 4b). The reason why the effect of miR-374b-5p was weak in CMM1 cells was considered to be that the decreased expression level of PTEN by transfection with miR-374b-5p in CMM1 cells was too high to attenuate the radiosensitivity.

On the contrary, transfection with miR-374b-5pi increased the expression level of PTEN in CMM1 and KMeC cells and decreased that of miR-374b in the cells tested (Figure 4c,d). Furthermore, miR-374b-5pi significantly increased the sensitivity of KMeC and KMeC/R cells to irradiation (Figure 4e). However, miR-374b-5pi did not have any effect on the radiosensitivity of CMM1 cells (Figure 4e). 

### 2.5. miR-374b-5p Directly Targeted PTEN

Because the results shown in Figure 4 indicated that miR-374b targeted PTEN, we performed a luciferase activity assay by using KMeC cells. We developed the plasmid vectors shown in Figure 5a. Expectedly, extrinsic miR-374b-5p successfully attenuated the luciferase activity of pMIR/PTEN, and the effect was significantly decreased in pMIR/mutPTEN (Figure 5b). These results indicated that miR-374b directly targeted PTEN in CoMM cells.

### 2.6. PTEN was Downregulated and miR-374b Upregulated in Recurrent Melanoma

To validate whether the results of the in vitro experiments reflected those of clinical samples, we examined the expression levels of PTEN and miR-374b in CoMM tissues obtained before radiation therapy and in the paired recurrent tissues after radiation therapy. All dogs achieved a partial response based on the response evaluation criteria for solid tumors in dogs (v1.0) [16] after radiation therapy. As shown in Figure 6a, the expression level of PTEN was significantly decreased (0.493 ± 0.296 vs. 0.358 ± 0.189 as the mean ± SD). On the other hand, that of miR-374b tended to be increased in the recurrent tissues compared with that in the primary tissues (0.218 ± 0.146 vs. 0.609 ± 0.396 as the mean ± SD; Figure 6b). 

## 3. Discussion

In this study, we validated the role of the miR-374b/PTEN signaling pathway in the radioresistance of CoMM cells. Our data showed that PTEN played a pivotal role in the radiosensitivity of CoMM cells and that its expression level was directly and negatively regulated by miR-374b. In addition, our data indicated that the expression levels of miR-374b and PTEN might be associated with radioresistance in a clinical specimen of CoMM, because these levels tended to differ between the samples obtained before radiation therapy and those taken after it.

The radioresistant CoMM cell line, KMeC/R, that we developed in this study had much less expression level of PTEN compared with the parental cell line. Moreover, the overexpression of PTEN restored the radiosensitivity of KMeC/R cells (Figure 2d). Furthermore, PTEN silencing successfully promoted the radioresistance of both CMM1 and KMeC cells (Figure 2d), although the ectopic expression of miR-374b affected the radiosensitivity of KMeC cells alone (Figure 4b). These data suggested that the role of PTEN is more critical than that of miR-374b in the radiosensitivity of CoMM cells.

The functions of miR-374b-5p in human cancer have been reported to be both oncogenic and anti-oncogenic [17,18,19]. In particular, miR-374b was shown to function as an oncogene by targeting PTEN, resulting in activation of the PI3K/Akt signaling cascade in human gastrointestinal stromal tumor cells [18]. In this study, it was indicated that miR-374b-5p directly targeted PTEN, which is one of the representative anti-oncogenes, in CoMM cells. While the ectopic expression of miR-374b-5p did not exhibit any effects on the cell growth of CoMM cells (data not shown), it was considered that miR-374b-5p possibly acted as an oncogene in CoMM cells due to the above reason. 

The effects of miR-374b-5p on the radioresistance of CMM1 cells were smaller than those of KMeC cells. We considered that the higher expression level of PTEN in CMM1 cells was associated with it. The inhibition of PTEN expression by the ectopic expression of miR-374b-5p might not be enough to promote the radioresistance of CMM1 cells. Transfection with miR-374b-5pi significantly promoted the radiosensitivity of KMeC/R cells without upregulating PTEN expression (Figure 4c). This result indicated that the mechanisms of downregulation of PTEN in KMeC/R cells may have been also associated with other causes such as deletion of or epigenetic changes in PTEN, in addition to miR-374b upregulation. Furthermore, it is also indicated that the mechanisms of radioresistance conferred by miR-374b include other target molecules. In cells originating from canine sources, the anti-oncogenic function of PTEN has been unclear. In a future study, more details about the function of PTEN and its differential expression need to be made and evaluated to validate its comprehensive roles in canine cancer.

The results shown in Figure 6 indicated that the expression level of PTEN was significantly decreased and that of miR-374b tended to be increased in recurrent melanoma tissues. These data indicated that the function of PTEN has a greater clinical impact for radiosensitivity of CoMM than that of miR-374b. We considered that these results correlated with the data obtained from the in vitro experiments, because the cancer cells included in the regrowth tissues after RT had possibly acquired the ability of radioresistance. However, unfortunately, the small sample set was a limitation in the current study, especially in analysis of miR-374b expression. Future analysis of the expression levels of PTEN and miR-374b by using a larger sample set would be desirable to validate the clinical impact of these levels associated with radioresistance. Furthermore, to validate the utility of the expression level of PTEN or miR-374b as a biomarker that predicts the efficacy of RT, correlation between the expression level of them and the degree of response to RT needs to be evaluated.

In conclusion, PTEN was found to be an important factor in the radiosensitivity of CoMM cells; and its expression was negatively controlled by miR-374b, whose effects on the radioresistance might vary according to cell types. Altogether, these factors are possibly promising markers of the radiosensitivity of CoMM. Canine melanoma has been considered as an important preclinical model of human melanoma [3,20]. Thus, the current study will possibly contribute to improved efficacy of RT for human melanoma.

## 4. Materials and Methods

### 4.1. Cell Culture and Cell Viability

CoMM cell lines CMM1 and KMeC were gifts from The University of Tokyo (Nakagawa), and cultured in RPMI1640 medium (Nacalai tesque, Kyoto, Japan), supplemented with 10% fetal bovine serum (Gibco, Waltham, MA, USA). Radioresistant KMeC/R cells, which were derived from KMeC cells, were developed by irradiation with a 4 MV linear accelerator (Primus mid energy, Canon medical systems, Tochigi, Japan) at a dose of 2 Gy every other day for 2 weeks. The human melanoma cell line A2058 was purchased from Health Science Research Resources Bank (Osaka, Japan). The cells were maintained in Dulbecco’s modified Eagle’s medium (Nacalai tesque, Kyoto, Japan). The number of viable cells was determined by performing the trypan blue dye exclusion test. 

### 4.2. Cell Count and Clonogenic Assay

To evaluate the number of viable cells, we seeded cells at 0.5 × 10^5^/well into 6-well plate the day before irradiation. At 72 h after irradiation, cell counts were performed. For the clonogenic assay, 500 cells were also introduced into 6-well plate. After 24 h, the cells were treated with a single dose of radiation (0, 1, 2 or 4 Gy). For the cells transfected with miRNA mimic or siRNA, the transfection was performed on the day after seeding and the cells were irradiated after 24 h of transfection. For irradiation, we used a 4 MV linear accelerator. The cells were incubated for 10 days after irradiation. Colonies were stained with crystal violet. Colonies containing >50 cells were counted under a microscope. The survival fraction was calculated as [number of colonies/number of cells plated] irradiated divided by [number of colonies/number of cells plated] non-irradiated.

### 4.3. Development of the PTEN Overexpressed CoMM Cells

The eukaryote pIRES-PTEN expression vector was generated by inserting the open reading frame of PTEN cDNA (accession number: XM_0225441428.1) into the EcoRI/BamHI site of the pIRESpuro3 vector (Clontech Laboratories, Inc., Mountain View, CA, USA). KMeC/R cells were seeded into a 25-cm^2^ flask and transfected with 1.0 μg/well of pIRESpuro3 as a control vector (mock) or pIRES-PTEN expression vector by using Lipofectamine 3000 (ThermoFisher Scientific, Waltham, MA, USA). Establishment of a PTEN stably expressed KMeC/R cell line (KMeCR/PTEN) and a control cell line (KMeCR/Mock) was achieved by selection using puromycin. 

### 4.4. Reagents and Antibodies

Short-interfering RNA (siRNA) for canine PTEN (5′-UAU AGG UCA AGU CCA AGU CGA ACC C -3′; siR-PTEN, Invitrogen, Carlsbad, CA, USA) was used for knockdown of PTEN. A miR-374b-5p mimic and a miR-374b-5p inhibitor (miR-374b-5pi) were purchased from ThermoFisher Scientific. Based on a database (miRBase; http://www.mirbase.org/) search, the sequence of canine miR-374b was the same as that of human miR-374b-5p. Transfection was achieved by using cationic liposomes, Lipofectamine RNAiMAX (Invitrogen, Carlsbad, CA, USA) according to the manufacturer’s Lipofection protocol. Pre-miR miRNA Precursor Molecules-Negative Control #2 (Applied Biosystems, Foster City, CA, USA) was used as a non-specific control miRNA. 

The following rabbit monoclonal antibody was used: Anti-PTEN (1:1000, Cell Signaling Technology, Denver, CO, USA). Horseradish peroxidase (HRP)-conjugated secondary antibodies were obtained from Cell Signaling. The loading control was prepared by re-incubating the same membrane with anti-β-actin antibody (Sigma, St. Louis, MO, USA).

### 4.5. Western Blotting

Total protein was extracted from whole of the cells by the procedure described previously [21]. Protein contents were measured with a DC Protein Assay Kit (Bio-Rad, Hercules, CA, USA). Ten micrograms of lysate protein for Western blotting was separated by SDS-PAGE using polyacrylamide gels and electroblotted onto a PVDF membrane (PerkinElmer Life Sciences, Boston, MA, USA). Details of the method used after blotting were described earlier [21]. The antibodies were properly diluted with TBS-T containing 2% bovine serum albumin and 0.01% sodium azide. 

### 4.6. MicroRNA Microarray and Quantitative RT-PCR (qRT-PCR) Using Real-Time PCR

Total RNA was isolated from cells by the phenol/guanidium thiocyanate method with DNase I treatment. microRNA microarray analysis was performed by using the GeneChip^TM^ miRNA Array (Filgen, Aichi, Japan). To determine the expression of miRNAs, we used TaqMan MicroRNA Assays (hsa-miR-374b-5p and RNU44; Thermo Fisher Scientific, Waltham, MA, USA). Real-time PCR was then performed by using a THUNDERBIRD^®^ Probe qPCR Mix (TOYOBO, Osaka, Japan). The relative expression level of miR-374b-5p was calculated by the ΔΔCt method. RNU44 was used as an internal control.

### 4.7. Luciferase Activity Assay

We constructed the sensor vector by joining the regions with a possible binding site from the 3′-UTR of canine PTEN to a luciferase reporter pMIR-control vector (Ambion, Foster City, CA, USA) to examine the target sequence recognized by miR-374b. Moreover, to generate the sensor vectors with mutations in the binding site for miR-374b, we mutated seed regions from TATTATA to GCAAGCT in region A and CGCATGT in region B (Figure 3A) by using a PrimeSTAR^®^ Mutagenesis Basal Kit (TaKaRa, Otsu, Japan). The sensor vector with mutations was submitted to Fasmac (Atsugi, Japan) for DNA sequencing. The cells were seeded in 12-well plates at a concentration of 0.5 × 10^5^/well the day before the transfection. The sensor vector (concentration; 1.0 μg/well) and 40 nM miR-374b-5p or non-specific control miRNA was used for the co-transfection of the cells by using cationic liposomes Lipofectamine RNAiMAX. Forty-eight hours after the co-transfection, luciferase activities were measured by using a Dual-Glo™ Luciferase Assay System (Promega, Madison, WI, USA) according to the manufacturer’s protocol. Firefly luciferase activity was normalized to Renilla luciferase activity.

### 4.8. Clinical Specimens

Fourteen CoMM tissues were obtained from 7 client-owned dogs with spontaneous oral melanoma, which animals had been biopsied for histological diagnosis before radiation therapy and cytoreductive surgery at recurrence after radiation therapy at the Veterinary Medical Center of Osaka Prefecture University. All tissues were pathologically diagnosed as malignant melanoma based on World Health Organization (WHO) criteria. In addition, of those, 6 dogs were classified stage III, and 1 dog was stage IV based on WHO staging scheme for dogs with oral melanoma [1]. Radiation therapy was performed by using a 4 MV linear accelerator (PRIMUS MID ENERGY, Canon Medical Systems, Tochigi, Japan). All dogs received the same radiation therapy protocol (8-Gy fractions administered once weekly for 4 weeks). The obtained tissues were used for protein and total RNA extraction. Informed consent in writing was obtained from each owner. Collection and distribution of the samples were approved by the appropriate institutional review board (admission number; R01-001).

### 4.9. Statistics

Each examination was performed in triplicate. All calculated data were compared by using the unpaired 2-tailed Student’s *t*-test, the paired 2-tailed Student’s *t*-test or one-way anova following Tukey methods. A *p*-value of less than 0.05 was considered to be statistically significant.

## Figures and Tables

**Figure 1 ijms-20-04631-f001:**
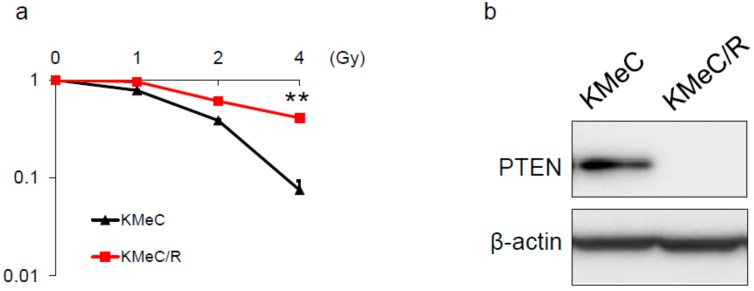
Establishment of the radioresistant KMeC cell line and the expression level of PTEN. (**a**) Results of colony formation assay using KMeC cells and their radioresistant ones. The assay was performed 14 days after the irradiation. (**b**) The protein expression level of PTEN in KMeC and KMeC/R cells. ** *p* < 0.01, for differences between KMeC and KMeC/R cells. Data are expressed as the mean + SD (*n* = 3).

**Figure 2 ijms-20-04631-f002:**
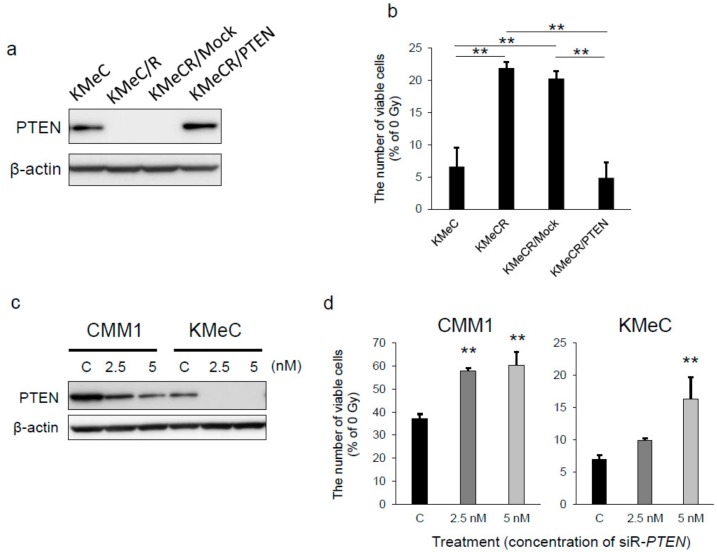
The effects of overexpression or silencing of PTEN on the radiosensitivity. (**a**) Successful overexpression of PTEN in KMeC/R cells. (**b**) Cell count analysis of the cells indicated after 8 Gy of irradiation. The cell count was performed 72 h after the irradiation. The data are given as a % of the number of the cells without irradiation. (**c**) Evaluation of the effects of silencing of PTEN by Western blotting analysis at 48 h after the transfection with siR-PTEN at indicated doses. (**d**) Cell count analysis of the cells transfected with siR-PTEN after 8 Gy of irradiation. The cell count was performed at 72 h after the irradiation. The data are shown as a % of the number of the cells without irradiation. ** *p* < 0.01, for differences between the samples indicated by the horizontal line or between the cells transfected with the negative control and siR-PTEN. Data are expressed as the mean + SD (*n* = 3).

**Figure 3 ijms-20-04631-f003:**
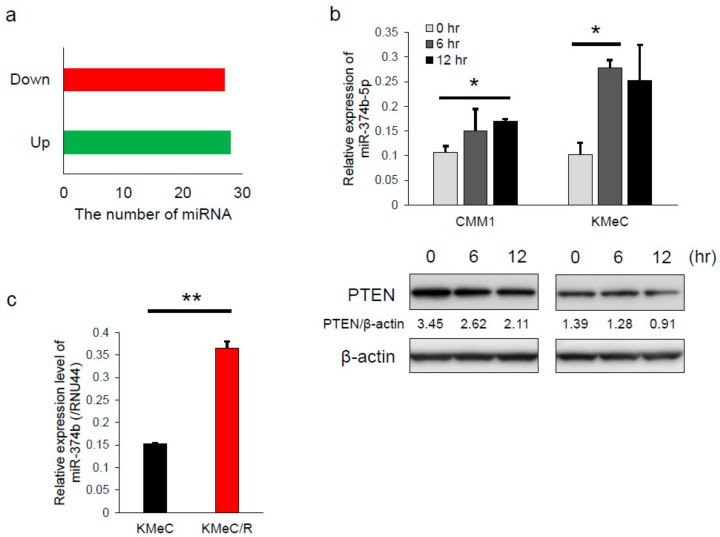
miR-374b upregulated in irradiated CoMM cells and radioresistant CoMM cells. (**a**) Results of microRNA microarray using human melanoma A2058 cells, showing numbers of upregulated and downregulated miRNAs. Total RNA was extracted before and 6 h after 2-Gy irradiation. (**b**) Expression level of miR-374b (**upper graph**) and PTEN (**lower panel**) in the CoMM cells before and 6 or 12 h after the 2-Gy irradiation. The values between the panel of PTEN and β-actin were the intensities of PTEN normalized to that of β-actin. (**c**) Expression level of miR-374b in KMeC cells and their radioresistant ones. * *p* < 0.05 and ** *p* < 0.01, for differences between the samples indicated by the horizontal lines. Data are expressed as the mean + SD (*n* = 3).

**Figure 4 ijms-20-04631-f004:**
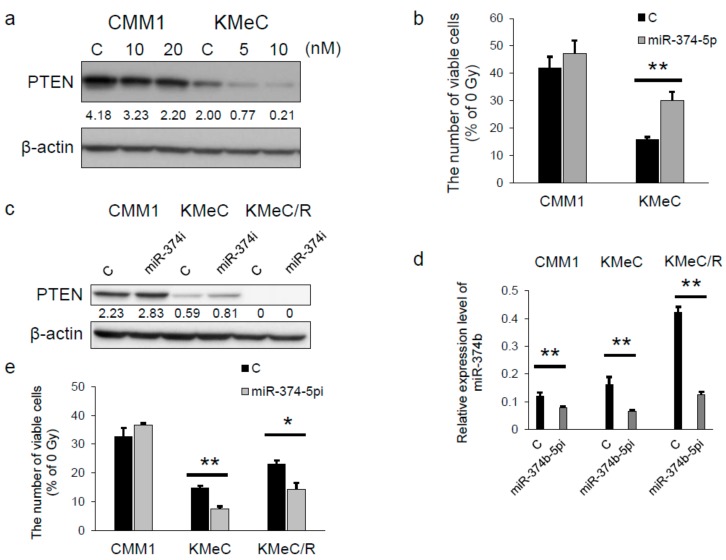
Ectopic expression of miR-374b-5p decreased the expression level of PTEN and conferred radioresistance on KMeC cells. (**a**) Western blot analysis of PTEN in the CMM1 and KMeC cells transfected with miR-374b-5p at the indicated doses. The assay was performed 48 h after the transfection. The values between the panel of PTEN and β-actin were the intensities of PTEN normalized to that of β-actin. (**b**) Cell count analysis of the cells transfected with miR-374b-5p (CMM1; 20 nM and KMeC; 10 nM) after 8 Gy of irradiation. The cell count was performed 72 h after the irradiation. Western blot analysis of PTEN (**c**) and the expression level of miR-374b (**d**) in the cells transfected with the negative control or 20 nM miR-374b-5pi. The assays were performed 72 h after the transfection. The values between the panel of PTEN and β-actin were the intensities of PTEN normalized to that of β-actin. (**e**) Cell count analysis of the cells transfected with miR-374b-5pi at a dose of 20 nM after 8 Gy of irradiation. The cell count was performed 72 h after the irradiation. The data are shown as a % of the number of the cells without irradiation. * *p* < 0.05 and ** *p* < 0.01, for differences between the samples indicated by the horizontal lines. Data are expressed as the mean + SD (*n* = 3).

**Figure 5 ijms-20-04631-f005:**
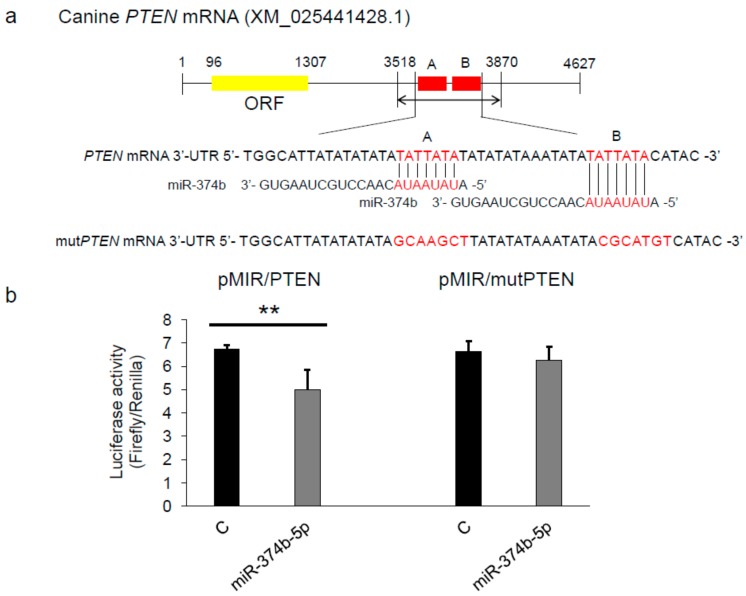
miR-374b-5p directly targeted PTEN. (**a**) Binding sites (A and B) of miR-374b in the 3′ UTR of canine PTEN mRNA. The wild-type binding sites were changed to the mutated ones as indicated. (**b**) Results of the luciferase activity assay using the pMIR reporter vector harboring the wild-type or mutated 3′ UTR of canine PTEN mRNA (the region of 3518 to 3870). The assay was performed 24 h after co-transfection with the reporter vector and miRNA. ** *p* < 0.01, for differences between the cells transfected with the negative control and miR-374b-5p. Data are expressed as the mean + SD (*n* = 3).

**Figure 6 ijms-20-04631-f006:**
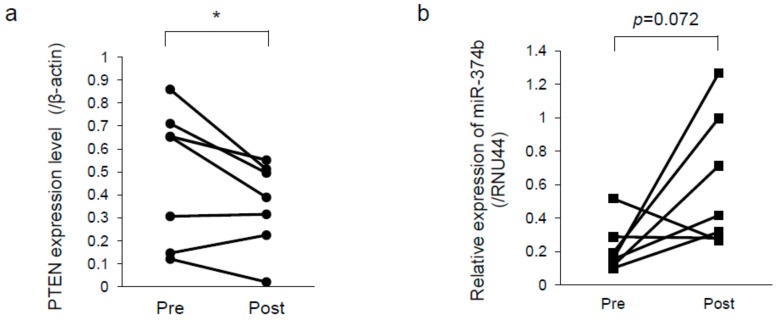
Expression levels of PTEN protein (**a**) and miR-374b (**b**) in the melanoma tissues that had spontaneously developed in the oral cavity of dogs. The tissues were harvested before radiation therapy and at recurrence. * *p* < 0.05, for differences between the bracketed samples.

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
