# Peer review of "PTEN, A Target of Microrna-374b, Contributes to the Radiosensitivity of Canine Oral Melanoma Cells"

_ijms, 2019, doi:10.3390/ijms20184631_

Round 1

Reviewer 1 Report

The manuscript written by Noguchi and coworkers describes the function of the tumor suppressor gene PTEN in radiotherapy of canine melanoma and its regulation by microRNA. Canine oral melanomas are aggressive tumors with a poor prognosis, which share pathogenic aspects with human melanomas and represent a preclinical model for human tumors. Using different melanoma cell lines (human and canine) the authors show that PTEN expression contributes to sensitivity to radiation in vitro, while silencing of PTEN increases radioresistence. Authors also show that PTEN is downregulated in recurrent melanoma in dogs following radiation therapy.

The manuscript is well written, easy to follow, and adequately illustrated. Used molecular techniques seem to be technically sound. The manuscript is of interest for scientist working the field of veterinary and human oncology.

I have only minor comments:

Line 44: the term “tumor suppressor gene” rather than “anti-oncogene” should be used. Line 63: correct spelling of “radioresistant” Line 154-161: Were oral melanomas classified based on WHO classification criteria. I assume all tumors in the present study represent malignant melanomas. This should be clearly stated in the text to avoid misunderstandings. Note that although the majority of oral melanomas in dogs are highly aggressive, a small subset of well-differentiated canine oral melanomas has been associated with prolonged survival times.

Author Response

Thank you for your critical comments. We did our best to response your grateful advisory comments. We marked the corrected portions in red.

Line 44: the term “tumor suppressor gene” rather than “anti-oncogene” should be used.

>As you advised, we corrected.

Line 63: correct spelling of “radioresistant”

>As you pointed out, we corrected.

Line 154-161: Were oral melanomas classified based on WHO classification criteria. I assume all tumors in the present study represent malignant melanomas. This should be clearly stated in the text to avoid misunderstandings. Note that although the majority of oral melanomas in dogs are highly aggressive, a small subset of well-differentiated canine oral melanomas has been associated with prolonged survival times.

>We added the statement that you suggested to section 4.8.

Reviewer 2 Report

This manuscript by Noguchi and colleagues examined how PTEN, a target of miR-374b may contribute to radiosensitivity of canine oral melanoma cells. Overall the results are interesting but overshadowed by the sloppiness of the authors with many many mistakes throughout the manuscript, suggesting careless and the absence of rigorous standards by the authors. 

In abstract, the statement of silencing of PTEN (line 22) significantly increased the radiosensitivity of CoMM cells tested. This statement is confusing, derivation of radioresistant CoMM celss (KMeC/R) showed downregulated PTEN, therefore silencing of PTEN should make CoMM cells more radioresistant not radiosensitive? What are CMM1 cells (Figure 2)?Why siRNA to PTEN has no effect on the PTEN expression in these cells? Downregulation of PTEN in CMM1 cells (Figure 3b) is not obvious, quantification of band intensities normalized to the loading control actin should be shown. How was the 6-hr timepoint selected as the time to examine miRNAs? Downregulated PTEN by miR-374b-5p in CMM1 is not convincing (line 115-116), quantification as suggested above should be done. Figure 4a mismatched of legend and figure, no AKT is shown, binding sites of miR-374b in the 3’UTR is not shown, no idea what miR-374-5pi is, no mention of it in the text at all. Results in figure 4C is not convincing at all. Not possible to judge results in figure 4c, d and e, without knowing exactly what miR-374-5pi is. How was the particular mutated PTEN selected? How many samples were evaluated in figure 6? 14? Different fonts in line 234. What is the rationale using A2058 to screen for miRNAs? A2058 has a mutated BRAF, do canine melanoma also carry this common mutation? How will this mutation affect the overall miRNAs?

Author Response

Thank you for your critical comments. We have done our best to respond to them. We marked the corrected portions in red.

In abstract, the statement of silencing of PTEN (line 22) significantly increased the radiosensitivity of CoMM cells tested. This statement is confusing, derivation of radioresistant CoMM celss (KMeC/R) showed downregulated PTEN, therefore silencing of PTEN should make CoMM cells more radioresistant not radiosensitive?

>We mistyped. We corrected.

What are CMM1 cells (Figure 2)? Why siRNA to PTEN has no effect on the PTEN expression in these cells? Downregulation of PTEN in CMM1 cells (Figure 3b) is not obvious, quantification of band intensities normalized to the loading control actin should be shown.

>CMM1 cells were CoMM cells. We considered that the effect of siRNA for PTEN was enough. Please refer to Fig. 2c and d. On the other hand, the degree of downregulation of PTEN by the treatment with irradiation in CMM1 cells was smaller than that in KMeC cells in Fig. 3b. We considered that this reason was due to the smaller degree of upregulation of miR-374b after irradiation in CMM1 cells than that of KMeC cells. As you suggested, we added the intensities to Fig. 3b.

How was the 6-hr timepoint selected as the time to examine miRNAs?

>It is known that the gene expression pattern starts to change within several hours after irradiation. Therefore, we examined microarray at 6 hr after irradiation.

Downregulated PTEN by miR-374b-5p in CMM1 is not convincing (line 115-116), quantification as suggested above should be done.

>As you suggested, we added the intensities to Fig. 4a.

Figure 4a mismatched of legend and figure, no AKT is shown, binding sites of miR-374b in the 3’UTR is not shown, no idea what miR-374-5pi is, no mention of it in the text at all.

>We mistook. The description was deleted in the legend of Fig. 4a.

Results in figure 4C is not convincing at all. Not possible to judge results in figure 4c, d and e, without knowing exactly what miR-374-5pi is.

>We added the intensities to Fig. 4c. As you pointed out, the effects of miR-374-5pi were week. We considered that the transfection rate of miR-374b-5pi might be too low. However, we judged that the data were worth including in the manuscript. If the data confuse you, we will decide to delete them.

How was the particular mutated PTEN selected?

>The sequences were decided to be inconsistent with the complementary sequences of the seed sequences of miR-374b. We randomly selected them.

How many samples were evaluated in figure 6? 14?

>Seven pretreatment tissues and paired 7 recurrent tissues were evaluated.

Different fonts in line 234.

>Sorry, we could not find the part you suggested.

What is the rationale using A2058 to screen for miRNAs? A2058 has a mutated BRAF, do canine melanoma also carry this common mutation? How will this mutation affect the overall miRNAs? 

>We considered that the microarray for canine might be more adequate. However, we used A2058 cells for microarray, because the number of probes on the platform for human was greater than that for dog. In fact, several genes of the miRNAs showing the altered expression level have not still remained to be recognized in the database for canine. These data are possibly useful in the future study.

BRAF mutation status in the cells used in this study has not been evaluated. However, it is known that BRAF mutation is not common in canine melanoma. As you considered, it has been reported that BRAF V600E mutation affects the expression level of several miRNAs other than miR-374b-5p.

Round 2

Reviewer 2 Report

The authors have addressed the concerns raised by the previous review. the different font on the words, irradiate and non-irradiate are now in line 235 and 236. Once these words have been corrected this manuscript is accepted for publication.

Author Response

Dear Reviewer 2,

Thank you for your kind comment. We identified the parts that you suggested and corrected font, because the font was subscript notation.

Sincerely yours,

Shunsuke Noguchi, D.V.M, Ph.D.